# Population Structure and Selection Signatures Underlying Domestication Inferred from Genome-Wide Copy Number Variations in Chinese Indigenous Pigs

**DOI:** 10.3390/genes13112026

**Published:** 2022-11-03

**Authors:** Wei Zhang, Mei Zhou, Linqing Liu, Shiguang Su, Lin Dong, Xinxin Meng, Xueting Li, Chonglong Wang

**Affiliations:** Key Laboratory of Pig Molecular Quantitative Genetics of Anhui Academy of Agricultural Sciences, Anhui Provincial Key Laboratory of Livestock and Poultry Product Safety Engineering, Institute of Animal Husbandry and Veterinary Medicine, Anhui Academy of Agricultural Sciences, Hefei 230031, China

**Keywords:** Wannan black pig, Asian wild boar, copy number variation (CNV), selection signatures, fixation index (F_ST_), whole genome resequencing

## Abstract

Single nucleotide polymorphism was widely used to perform genetic and evolution research in pigs. However, little is known about the effect of copy number variation (CNV) on characteristics in pigs. This study performed a genome-wide comparison of CNVs between Wannan black pigs (WBP) and Asian wild boars (AWB), using whole genome resequencing data. By using Manta, we detected in total 28,720 CNVs that covered approximately 1.98% of the pig genome length. We identified 288 selected CNVs (top 1%) by performing Fst statistics. Functional enrichment analyses for genes located in selected CNVs were found to be muscle related (*NDN*, *TMOD4*, *SFRP1*, and *SMYD3*), reproduction related (*GJA1*, *CYP26B1*, *WNT5A*, *SRD5A2*, *PTPN11*, *SPEF2*, and *CCNB1*), residual feed intake (RFI) related (*MAP3K5*), and ear size related (*WIF1*). This study provides essential information on selected CNVs in Wannan black pigs for further research on the genetic basis of the complex phenotypic and provides essential information for direction in the protection and utilization of Wannan black pig.

## 1. Introduction

Pigs (Sus scrofa) are one of the important agricultural animals, originating from the Anatolia and Mekong valley about 9000 years before the present [1,2]. Pigs play an important role in the transition of human development and evolution, from nomadic to contemporary civilization by means of providing meat, organic fertilizer, raw material for chemical industry, and a medical model for human disease. The long-term process of the natural and artificial selection of wild boars has led to striking morphological and behavioral changes and has resulted in modern, abundant pig breeds, which have captured the interests of animal biologists generation after generation to elucidate the genetic mechanism of the domestication and further to protect and utilize the excellent germplasm resources.

The development of genetics and genomic technologies as well as the decline in the sequencing cost provide more powerful tools for a comprehensive understanding of the genetic architectures and evolutionary trajectories of complicated traits in pigs. The genes related to domestication in pigs have been widely discovered, such as body-length-related genes [3,4], coat-color-related genes [5,6,7], high-altitude-related genes [8], and reproduction-related genes [9]. However, the genes characterized are disproportionately skewed toward single nucleotide polymorphisms (SNPs). Another type of variations, copy number variation (CNV), has been neglected and little known in domestication. CNV can be defined as DNA segments ranging in size from 50 base pairs (bp) to several megabases (Mb) in which insertion, duplication or deletion events have occurred [10,11,12]. Compared with the effect of widely studied SNPs, CNVS have significant genomic effects, including direct effects on gene dosage, indirect changes in gene expression through positional effects, revelation of recessive alleles or regulatory polymorphisms, loss of regulatory elements and influence on the evolution of new genes [13]. In humans, the CNV of the AMY1 (amylase α 1) was found to be associated with starch, indicating the different cultural change in different human population [11]. A 163 bp deletion in MKL1 (megakaryoblastic leukemia (translocation) 1) was found to be related to high-altitude adaption in the Tibetan human population [14]. In cattle, the CNV of the CATHL4 (cathelicidin 4) and ULBP17 (UL16-binding protein 17) could explain the pathogen and parasite resistance of the Nelore cattle [15]. In dogs, an insertion of the AKR1B1 (aldo-keto reductase family 1 member B) was associated with fatty acid synthesis and antioxidant ability, and it explains the dietary shifts during the agricultural revolution [16]. In pigs, a CNV of the MSRB3 (methionine sulfoxide reductase B3) was shown to increase porcine ear size [17]. CNV of the MTHFSD (methenyltetrahydrofolate synthetase domain containing) affects litter size in the Chinese indigenous Xiang pig [18]. Considering the vital role of CNV in humans and animals, excavating more CNVs associated with important traits and elucidating the genetic mechanism of pig domestication is desired.

The Wannan black pig (WBP) is a typical Chinese-native disease-resistant breed with high fertility, excellent meat quality, good maternal stability, and a crude-feed tolerance that is mainly found in the south regions of Anhui Province, China. WBP is favored by people in the Yangtze River Delta region. Meanwhile, it is the best raw material for the famous “Huizhou ham” and Anhui cuisine “braised pork”. In 1982, the purebred female of the WBP reached a population size of 1702. However, with the rapid expansion of commercial pigs, the lack of effective conservation efforts and the effect of African Swine Fever, WBPs are facing a dramatic decline in population size and loss in genetic characteristics. In 2019, the number of purebred females decreased to 360. As a small indigenous population, WBP are at risk of extinction. In our previous studies, reproduction-, lipid-, meat-, and immune-related genes were identified in WBP, which could explain a part of the characteristics of WBP [19,20,21,22,23]. However, the above studies are all focused on the effect of SNP in revealing the genetic mechanism of characteristics in WBP. Considering the vital effect of CNVs in the domestication and elucidating the genetic mechanism of complex phenotypes, it is necessary to detect the CNVs in WBP and investigate the selection signatures of WBP compared with the Asian wild boar (AWB) to uncover its characters.

The purpose of this study is (1) to perform a genome-wide CNV analysis in WBP and AWB, including 45 WBPs and 19 AWBs, using whole-genome-resequencing data; (2) to reveal the differences in population structure of WBP and AWB based on CNV; (3) to identify the candidate-selected CNVs by calculating the FST through WBP and AWB and compare the candidate-selected CNVs with economically important traits in WBP. The information gained in this study will be a valuable resource for understanding the role of CNVs in pig evolution and breed formation and will also provide new insight into the protection and utilization of WBP.

## 2. Materials and Methods

### 2.1. Ethics Statement

All animal work was carried out according to the approved guidelines established by the Ministry of Agriculture of China. This study was conducted in accordance with and was approved by the Animal Care Committee of the Anhui Academy of Agricultural Sciences (Hefei, China; no. AAAS2020-04).

### 2.2. Blood Samples and Resequencing

Samples were collected from blood samples of 25 Wannan black pigs. Briefly, these individuals were sampled from the nucleus population of Wannan black pig in a conservation farm in Jixi County, Anhui Province, China. Genomic DNA was extracted using the standard phenol–chloroform method [24]. The DNA quality was measured by Nanodrop spectrophotometer (Thermo Fisher Scientific, Waltham, MA, USA) and 0.5% agarose gel. Subsequently, all samples were constructed from a library (paired end, 2 × 150 bp) and sequenced using an Illumina NovaSeq 6000 platform (Illumina, San Diego, CA, USA) through the Novogene service (Beijing, China). In addition, the genomic data of 20 Wannan black pig and 6 AWB in our previous report were used with accession number PRJNA524263 and PRJNA699491, respectively [19,25]. The genomic data of 13 AWB from the National Center for Biotechnology Information (NCBI) were downloaded with accession number PRJNA213179, PRJNA186497 and PRJEB1683, respectively [8,26,27]. In total, there are 45 Wannan black pig samples, and 19 AWB samples used in this study.

### 2.3. Read Mapping and CNV Calling

In the processing of CNV detection, adapters and low-quality reads were removed using the NGSQC Toolkit (v.2.30) [28]. The filtered reads were then aligned to the pig reference genome (ftp://ftp.ensembl.org/pub/release-99/variation/gvf/sus_scrofa/, accessed on 10 September 2022) by the Burrows–Wheeler aligner (BWA) with the default parameter. The CNVs were detected following four steps. Firstly, Manta [29] was used to obtain the CNV results of each individual. Secondly, Paragraph [30] was then used to genotype the variants of each individual. Thirdly, de-redundancy at the group level was performed based on the information of location (deletion is 50% overlapped; ins, and dup are set to overlap 90%) and genotyping results (population typing consistency ≥ 0.95). Fourthly, the quality controls were under the following criterion: ABS(INFO/SVLEN) ≤ 10,000,000, INFO/ExcHet ≥ 0.05, F_MISSING ≤ 0.2, and INFO/MAF > 0. To better understand the frequencies of CNV in different populations, the plink software was used to statistically measure the frequency in both populations, which was visualized using R (v4.2.0).

### 2.4. Population Genetic Structure Analysis

To infer the population structure of WBP and AWB pig using the detected CNVs, the VCF file of CNV was converted to the PLINK input file formats (map and. ped) by PLINK software v.1.90 [31]. The principal component analysis was performed using the SMARTPCA program implemented in EIGENSOFT [32]. Additionally, phylogenetic trees were inferred through the neighbor-joining (NJ) method and implemented in MEGA and ITOL (https://itol.embl.de/, accessed on 16 September 2022) for visualization. The population structure was deduced using the ADMIXTURE software v1.3.0 [33].

### 2.5. Identification of Selection Signatures

The statistical measure fixation index (FST) was used to explore population-differential CNVs between WBP and AWB across the whole genome (not considering sex chromosomes). The formula for Fst calculation is Fst = (Ht − Hs)/Ht, where Ht is the expected heterozygosity of the population, and Hs is the expected heterozygosity of the subgroup. The FST is based on population differentiation and was first defined by Lewontin and Krakauer [34] based on coefficient F [35]. It was developed by Weir and Cockerham [36], Akey et al. [37], and Gianola et al. [38]. The top 1% of the FST value was selected as the threshold, and the selected signatures were obtained by screening. Finally, the selected signatures were annotated, and the candidate genes were analyzed for functional enrichment of KEGG and GO pathways by using the ClusterProfiler (version v3.14.0) [39] in R (version 3.6.1). The terms and pathways exhibiting *p*-values < 0.05 were considered significant.

## 3. Results

### 3.1. Copy Number Variation Identification in Wannan Black Pig

To detect genome-wide CNV in WBP, we performed whole-genome sequencing of 25 unrelated WBPs, which yielded 835.47 Gb data with an average depth of 11.57 (Appendix A). The previous sequenced and downloaded data were 1164.24 Gb data with an average depth of 11.50 (Appendix A). A total of 1999.23 Gb data were used in this study. A total of 15,972 CNVs covering ~26.08 Mb (1.15% of the pig genome) were identified in WBP, including 12,082 Del, 418 Dup, and 3472 Ins (Table 1). A total of 20,774 CNVs covering ~25.76 Mb (1.15% of the pig genome) were detected in AWB, including 13,518 Del, 783 Dup, and 6473 Ins (Table 1). The Del was greatest in WBP and AWB, covering 75.64% and 65.07, separately. The average and median length of CNV in WBP are 1638 bp and 278 bp, separately (Appendix A). The average and median length of CNV in AWB are 1240 bp and 132 bp. We can see that the number of CNV in WBP is less than the AWB, but the average and median lengths of WBP are much greater than the AWB, which is maybe caused by the selection.

After merging the CNVs based on the criteria in the Materials and Method section, a total of 28,720 high-quality CNVs covering 49.96 Mb (1.98% of the pig genome, Appendix A) were determined. The Venn diagram revealed that the WBP and AWB had 8026 CNVs (27.95%, 28,720 total) in common, and 27.62% CNVs were unique to WBP, whereas 44.43% were unique to AWB (Figure 1A). For analysis of the frequency of the merged CNVs in two population, we divided the frequency into 10 groups (0–0.1, 0.1–0.2, 0.2–0.3, 0.3–0.4, 0.4–0.5, 0.5–0.6, 0.6–0.7, 0.7–0.8, 0.8–0.9, and 0.9–1; see Appendix A). We can see from Figure 1B,C that the frequency of 0–0.1 was greatest in two populations. With the increase infrequency in del, the number first decreased and then increased, and dup was decreased with the increase in frequency. The trend of ins was similar to that of del. In order to fully understand the distribution of the merged CNVs in the genes, annotation of the total CNVs were conducted and revealed that they were most abundant in the intergenic regions (41.49%) and intronic regions (33.29%), followed by the exonic region (19.07%), downstream (0.9%), upstream (0.65%), and splicing (0.32%) (Table 2).

### 3.2. Phylogenic Construction, Principal Component Analysis, and Admixture Analyses

To explore the relationship between WBP and AWB populations, we firstly examined the neighbor-joining (NJ) tree, and the result revealed that the AWB and WBP were clustered separately (Figure 2A). Secondly, the PCA were conducted using the first two PC, and PC1 and PC2 explained 13.77% and 6.87% of the total variations, respectively (Figure 2B). Lastly, to further understand the degree of mixture in the two populations, K = 2 was used. As shown in Figure 2C, it can separate all WBPs from AWBs.

### 3.3. Identification of Selection Signatures Based on CNVs

We performed the F_ST_ method to screen for the potential CNVs under selection in the genome of WBP. A total of 288 CNVs were identified as selected based on the top 1% threshold of the F_ST_ (threshold: 1%, F_ST_ = 0.71, Figure 3, Appendix A), which harbors 199 genes (Appendix A). In order to assess the function of these genes, GO terms and KEGG pathway analyses were conducted. In the Go analysis, a total of 109 terms were significantly enriched (Appendix A), and these genes were related to growth (GO:0040007, *p* = 0.0245, 9 genes), reproduction (GO:0000003, *p* = 0.03215, 12 genes), digestive system development (GO:0055123, *p* = 0.02462, 4 genes), and limb morphogenesis (GO:0035108, *p* = 0.01521, 5 genes) (Figure 4A). In the KEGG analysis, there are five pathways that were significantly enriched (Appendix A), including the MAPK signaling pathway (ssc04010, *p* = 0.04517, 10 genes), Ras signaling pathway (ssc04014, *p* = 0.04517, 9 genes), and nitrogen metabolism (ssc00910, *p* = 0.04318, 3 genes) (Figure 4B).

## 4. Discussion

The resource of pig breeds is an important strategic resource to ensure the safety and sustainable development of the pig industry. Pigs play an important role in China’s national economy: (i) the net pig meat production of 2021 reached 52.96 million tons; (ii) the pig industry provides 70 million jobs; (iii) the output value of pork is as much as CNY 1.5 trillion. Since many Chinese native pig breeds are facing the reduction in population number, we should explore the germplasm characteristics of different local pigs at a deeper level. Then, carrying out scientific breeding to improve the competitiveness will be a new way for the development of local pig and industrialization. In this study, we performed the whole-genome sequencing of 25 unrelated WBP, combined them with the previous sequenced data, and downloaded data jointly to conduct the population structure and selection signatures underlying domestication based on CNV. Manta was effective in identifying the CNV in previous studies. Previous studies in humans with about 69 detection algorithms for SV revealed that Manta is better than most software [40]. Pourya et al. found characteristics-related genes in mink by using Manta [41]. A total of 15,972 and 20,774 CNVs were identified in WBP and AWB, separately. Although the number of CNV in WBP is less than the AWB, the average and median lengths of WBP are much greater, which may result from the selection. A total of 288 CNVs were identified as selected, and 199 genes were harbored in the selected CNVs. Functional enrichment analysis revealed that the genes were related to growth, reproduction, digestive system development, limb morphogenesis, MAPK signaling pathway, and Ras signaling pathway.

Several genes were found related to the regulation of muscle. Necdin (*NDN*), a member of the melanoma antigen (MAGE) family, was identified as a paternally imprinted gene in newborn piglets and could regulate the growth of muscle [42]. Through functional analysis of the position of the CNV in NDN with PigQTLdb (https://www.animalgenome.org/cgi-bin/QTLdb/SS/index, accessed on 13 October 2022), eight QTLs were identified, including fat-to-meat ratio (ID = 5677, 12,742), lion muscle depth (ID = 29,579), and lion muscle area (ID = 2742, 3640, 3796, 38,072). Tropomodulin 4 (*Tmod4*), a member of the Tmod family, plays an important role in thin filament length regulation and myofibril assembly. The Tmod4 gene was found to serve as a switch between myogenesis and adipogenesis, which resulted in the balanced development between skeletal muscle and adipose tissue in pig [43]. Through functional analysis of the position of the CNV in Tmod4 with PigQTLdb, four QTLs were identified, including the diameter of type IIb muscle fibers (ID = 2807, 2810), number of muscle fibers per unit area (ID = 2808), and diameter of muscle fibers (ID = 2809). Secreted frizzled-related protein 1 (*SFRP1*), a member of the SFRP family that inhibits Wnt signaling [44], could regulate the skeletal muscle development of the pig in embryonic stages [45]. Through functional analysis of the position of the CNV in SFRP1 with PigQTLdb, three QTLs were identified, including drip loss (ID = 12,072), and shear force (ID = 3016, 12,075). SET and MYND domain-containing protein 3 (*SMYD3*) is a member of the SMYD family that plays a vital role in the myofibril assembly of skeletal and cardiac muscles [46]. It is necessary for regulating skeletal muscle and myocardial development [47]. Through functional analysis of the position of the CNV in SMYD3 with PigQTLdb, three QTLs were identified, including percentage type I fibers (ID = 7012, 7026), and percentage type IIa fibers (ID = 7034).

Some genes in the selected CNVs were found to be reproduction related. Gap junction protein α 1 (*GJA1*), also known as CX43, is a component of gap junctions. It was significantly enriched in embryo development (GO:0009790, *p* = 0.01649). The knockdown of GJA1 in pigs results in a significant reduction in the blastocyst development rate and the total number of cells in the blastocysts, and induces autophagy and apoptosis, which imply its vital role for the development and preimplantation of porcine embryos [48]. It regulates oocyte meiosis resumption, and lower levels of GJA1 in cumulus cells are beneficial for oocyte maturation [49]. Cytochrome P450 family 26 subfamily B member 1 (*CYP26B1*) is necessary for embryonic development and survival during fetal life [50]. Through functional analysis of the position of the CNV in CYP26B1 with PigQTLdb, two QTLs were identified, including corpus luteum number (ID = 51,518,044). Cyp26b1-null mice exhibit pronounced skeletal abnormalities that are characterized by either underossification or the loss of endochondral and intramembranous-derived bones [51]. In this study, Wnt family member 5A (*WNT5A*) is involved in the Wnt signaling pathway, which is related to spermatogenesis, epididymal sperm maturation, and embryonic sexual development [52,53,54,55]. WNT5A was proved to be a key regulator of follicle development and gonadotropin responsiveness [56,57]. The vertebrate limb is a classical model for understanding the patterning of three-dimensional structures during embryonic development. Previous research elucidated that the WNT5A plays a necessary role in the proper orientation of cell movements and cell division, which sheds light on the cellular basis of vertebrate limb bud morphogenesis [58]. Through functional analysis of the position of the CNV in WNT5A with PigQTLdb, two QTLs were identified, including the corpus luteum number (ID = 4,938,826). Secreted frizzled related protein 1 (*SFRP1*) was also found to be selected, and it was able to regulate spermatid adhesion as well as their release during spermiation in the testes [59]. A mouse knock-out model showed malformation in the development of the testes and impaired maturation of the reproductive tract [55]. Steroid 5 α-reductase 2 (*SRD5A2*) was found to be associated with fertility in humans and animals. In humans, the variation in SRD5A2 was significantly correlated with semen quality [60]. In cattle, it was found to be selected in Jiaxian Red cattle and associated with fertility [61]. In goat, genomic analysis revealed that it was significantly enriched in steroid hormone biosynthesis, which was related to reproduction [62]. The protein tyrosine phosphatase non-receptor 11 (*PTPN11*) gene encodes for a Src homology-2 domain-containing protein tyrosine phosphatase 2 (SHP2). Shp2 plays an indispensable role in spermatogenesis by mediating follicle-stimulating hormone (FSH) and testosterone signals [63,64,65,66]. Meanwhile, it also affects energy balance and lipid and glucose metabolisms [67]. Additionally, in a study with mice, SHP2 was reported to be associated with obesity [68]. Sperm flagellar 2 (*SPEF2*) is mainly expressed in the sperm flagellum and plays a crucial role in sperm tail development [69]. Dysfunctional mutations in SPEF2 impair sperm motility and cause a short-tail phenotype in human and animals [69,70,71,72]. Cyclin B1 (*CCNB1*), an important regulator in cell cycle machinery, is proved essential for mouse embryonic development; those lacking CCNB1 were unable to proliferate normally, and apoptosis increased [73].

Some genes associated with other economic traits were also found in this study. Residual feed intake (RFI) is an important quantitative trait in the pig industry [74,75,76,77]. Understanding the genetic basis underlying this complex trait will help in the efficient selection of pigs, thereby being beneficial for the pig producers. Mitogen-activated protein kinase kinase kinase 5 (*MAP3K5*) was found to be selected in this study, and it is significantly enriched the MAPK signaling pathway. MAPK signaling pathway is an important signal transduction system that mediates the extracellular signal to an intracellular response in eukaryotic cells. Serão et al. (2013) [78] found that some genes in the MAPK signaling pathway were associated with RFI [78]. The MAP3K5 gene was found as a selected gene associated with RFI in African Ankole cattle and the shorthorn Muturu cattle [79,80]. In pigs, it was also found that there was an association with RFI by association analysis [81,82]. A deletion in the intronic region was found (start = 27,584,917 bp, end = 27,585,183 bp). Through functional analysis of the position of the CNV in MAP3K5 with PigQTLdb, we found three QTLs related to RFI, including average daily gain (ID= 319, 5680) and feed-conversion ratio (ID = 29,552). The ear size of pigs has been selectively bred by humans in many areas of China for a long time, and as a result, most Chinese pig breeds have medium- to large-sized ears [83]. Ear size was regarded as an important characteristic distinguishing different pig breeds [84]. WNT inhibitory factor 1 (*WIF1*) was found to be the selected gene in this study and also identified as the selected gene in previous studies in pigs and cattle [85,86]. WIF1 was found to be able to regulate the ear size of pigs and dogs based on association analysis [85,87]. There is a deletion in the WIF1 gene (start = 29,516,333 bp, end = 29,516,804 bp). Through functional analysis of the position of the CNV in WIF1 with PigQTLdb, two QTLs were identified, including ear erectness (ID = 56,475,652).

Although some interesting findings were reported here, the limitation of this study should not be neglected. On the one hand, the genes in the selected CNVs were confirmed to be associated with important traits in previous research studies, which were mainly based on the SNPs. There is little reference about the effect of CNV on these important traits. It is also fortunate for us that the role of the genes was clear, and CNVs have significant genomic effects that regulate the expression of the genes. On the other hand, although we have obtained the genotype data of the WBP, the phenotype data for the important economic traits are missing. The concrete function of the CNVs is still obscure. The limitations might impact the observation of this study and should be overcome in further investigations by (i) enlarging the study population and collecting samples and phenotypes, and (ii) verifying the effect of CNVs in different traits by association analyses.

## 5. Conclusions

In this study, we first detected the CNVs in the WBP population and AWB population. Secondly, the population structure of WBP and AWB based on CNV was conducted. Thirdly, the selection signatures in WBP were identified with Fst based on CNV, and it was found that several selected CNVs are associated with muscle-related, reproduction-related, RFI, and ear-size characteristics. These findings expand our knowledge on the effect of CNV in important economic traits in pigs and provide a valuable resource for future genetic association analysis to improve the genome-assisted breeding of pigs.

## Figures and Tables

**Figure 1 genes-13-02026-f001:**
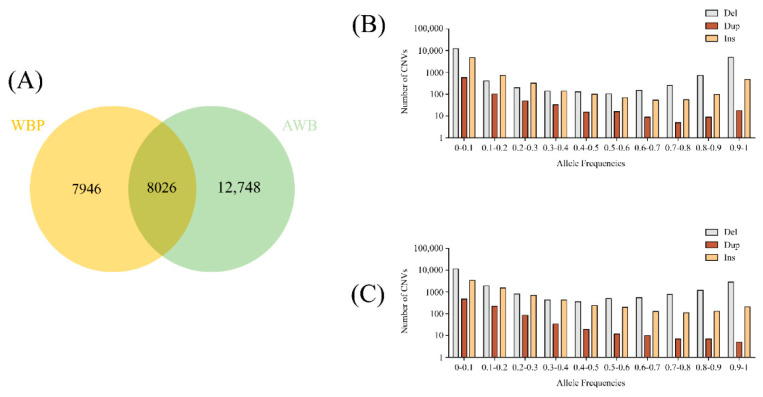
(**A**)The Venn diagram of CNVs between WBP and AWB. (**B**) The allele frequencies of variants in the WBP (*n* = 45). (**C**) The allele frequencies of variants in the AWB (*n* = 19).

**Figure 2 genes-13-02026-f002:**
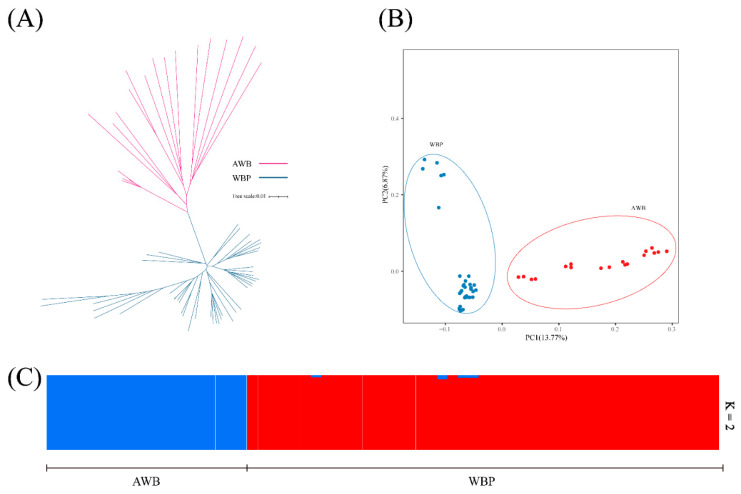
(**A**) Neighbor-joining tree constructed from CNV data in study population. (**B**) PCA plots for the first two PCs for all 64 individuals, PC1 and PC2 explain 13.77% and 6.87% of the total variations, respectively. (**C**) Structure analysis on all the AWB and WBP with K = 2.

**Figure 3 genes-13-02026-f003:**
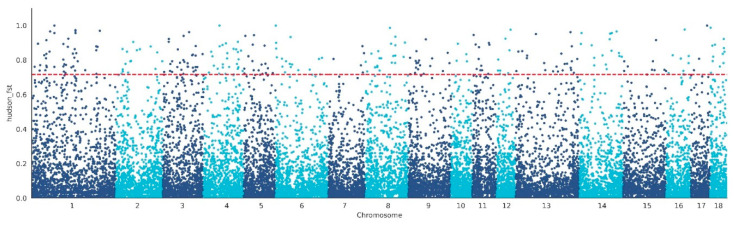
Identification of the selected CNVs in WBP with Fst. The red line represents the top 1% threshold.

**Figure 4 genes-13-02026-f004:**
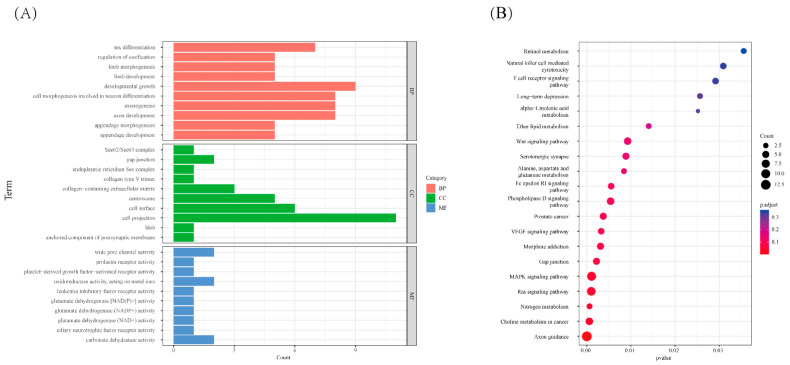
The function analysis of the selected genes, (**A**) GO analysis of the selected genes, red referring to biological process, green referring to cellular component and blue referring to molecular function. (**B**) KEGG analysis of the selected genes, the size of dot refers to the number genes related to pathway, and the red to blue indicate the significant *p*-value change.

**Table 1 genes-13-02026-t001:** The statistic of CNV in WBP and AWB.

Population	Total Number	Number of Variants			Total Length(bp)/ Genome Ratio	Length (bp)/Genome Ratio		
		del	dup	ins		del	dup	ins
AWB	20,774	13,518	783	6473	25,759,359/1.14%	24,872,366/1.1%	53,959/0.0024%	833,034/0.0376%
WBP	15,972	12,082	418	3472	26,087,337/1.15%	25,614,866/1.13%	26,398/0.0012%	446,073/0.019%

**Table 2 genes-13-02026-t002:** Annotation of the merged CNVs.

Classification	No. of Variants
Downstream	261
Upstream	188
Downstream; upstream	12
Exonic	5478
Intron	9562
Intergenic	11,918
Splicing	93
ncRNA	983
UTR3	166
UTR5	57
UTR3; UTR5	2

## Data Availability

The datasets presented in this study can be found in online repositories. The names of the repository/repositories and accession number(s) can be found in the article. The dataset used and analyzed during the current study is available from the corresponding author on reasonable request.

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
