# Peer review of "Population Structure and Selection Signatures Underlying Domestication Inferred from Genome-Wide Copy Number Variations in Chinese Indigenous Pigs"

_genes, 2022, doi:10.3390/genes13112026_

Round 1

Reviewer 1 Report

  1. Line 114-115. Did you use the CNV regions based on the overlap? I don’t completely understand the difference between step 2 and step 4. Perhaps the authors could elaborate on the specifics of these two steps?
  2. Line 125. What input files are used here? (Output plink?) I believe, this clarification will make it easier for the reader to follow the process in the article.
  3. Line 131. It is ambiguous how the FST was calculated. Would the authors be able to explain this procedure in more detail in the materials and methods (how were the CNVs coded?)
  4. Line 162. Did I understand correctly, that the division into groups is based on the second step 2.3 part?
  5. Line 169. As a reader I did not fully understand the meaning of Table 2. Are these CNVs common to the two species? Could the authors explain the meaning of Table 2 statistics in this study, because the task is based on finding breed-specific CNVs?

Author Response

Response to Reviewer 1 Comments

Point 1: Line 114-115. Did you use the CNV regions based on the overlap? I don’t completely understand the difference between step 2 and step 4. Perhaps the authors could elaborate on the specifics of these two steps?

Response 1: Thank you very much for your valuable suggestion. It is quite right that the description of calling CNV are confounding, especially the step 2 and step 4. I have removed the step 2 for better reading. The CNV calling process were marked in red in the line 113 to 120.

Point 2: Line 125. What input files are used here? (Output plink?) I believe, this clarification will make it easier for the reader to follow the process in the article.

Response 2: Thank you very much for your constructive suggestion. The orgin description is indeed fuzzy for understanding. I have added the detail information for the process and marked with red in line 124 to line 126 in the manuscript and provided the reference.

Point 3: Line 131. It is ambiguous how the FST was calculated. Would the authors be able to explain this procedure in more detail in the materials and methods (how were the CNVs coded?)

Response 3: Thank you for your suggestion. I have added the how the FST was calculated and marked red in the manuscript in line 134 to line 135.

Point 4: Line 162. Did I understand correctly, that the division into groups is based on the second step 2.3 part?

Response 4: Thank you for your suggestion. I am sorry for the unclear statement of the information about the division into group. I have adde “the merged CNVs” in line 164 and marked in red.

Point 5: Line 169. As a reader I did not fully understand the meaning of Table 2. Are these CNVs common to the two species? Could the authors explain the meaning of Table 2 statistics in this study, because the task is based on finding breed-specific CNVs?

Response 5: Thank you for your suggestion. It is important that a article should readable for all people and have a clear description. The orgin thought is to clarify where the merged CNVs are distributed in the gene. We have added detail information in line 169 to line 170.

Once again, I would like to express my heartfelt thanks to you for revising our paper, and thank you for your constructive suggestions, which have greatly improved the manuscript comprehensively.

Reviewer 2 Report

The manuscript by Zhang et al. displays a study in Wannan black pig. It aims at detecting CNVs from whole genome sequencing data in order to learn about population structure and signatures of selection in this pig breed. This study is interesting and original. However, I am concerned about the applied methods for CNV detection. If CNVs are discussed, the authors should not only consider single deletions, insertions or duplications but also the variation in copy numbers. This variation with regard to CNV-count is very well known to have an important impact on development or disease (for example on body size or cerebellar diseases among others). The tool Manta used in this study was originally designed to target rare structural germline variants, but it is not an appropriate tool to detect the whole range of copy number variants. I suggest to perform an additional analysis using a CNV detection tool (e.g. PennCNV, QuantiSNP). Copy numerbs should be associated with traits of interest, as well as discussed in introduction and discussion.

Furthermore, the authors should consider validating some CNVs of interest using qPCR as proof of reliable data analysis. The matter of validation of data analysis should also be considered in the discussion.

Minor comments:

AWB: explain abbreviation at first use

Use consistently among text and figures either “AWB” or “Wild” or else

Line 189: terms

Discussion: First paragraph seems to be a leftover from the template. Remove.

Author Response

Response to Reviewer 2 Comments

Point 1: The manuscript by Zhang et al. displays a study in Wannan black pig. It aims at detecting CNVs from whole genome sequencing data in order to learn about population structure and signatures of selection in this pig breed. This study is interesting and original. However, I am concerned about the applied methods for CNV detection. If CNVs are discussed, the authors should not only consider single deletions, insertions or duplications but also the variation in copy numbers. This variation with regard to CNV-count is very well known to have an important impact on development or disease (for example on body size or cerebellar diseases among others). The tool Manta used in this study was originally designed to target rare structural germline variants, but it is not an appropriate tool to detect the whole range of copy number variants. I suggest to perform an additional analysis using a CNV detection tool (e.g. PennCNV, QuantiSNP). Copy numerbs should be associated with traits of interest, as well as discussed in introduction and discussion.

Response 1: It is right for your valuable suggestion. Identification CNVs in pig is just a first step. What is important is that how the CNV regulate the economic traits. That is to say, how the copy number affect the phenotype variation. The problems we faced are the dual impact of COVID-19 and swine fever. It is particularly difficult to collect accurate phenotypic data, which result in that it is difficult to study the specific regulatory impact of copy number changes on economic traits. In the next step, we will conduct the association analysis between copy number changes and phenotypic traits, in order to reveal the specific regulatory mechanism.

It is right as your valuable suggestion that more tools should be used to call CNV. Meanwhile, the manta was effective in identifying CNVs in many studies, such as human and mink. We think the results in this manuscript maybe could explain some questions about the domestication and provide important information for better utilization of the Wannan black pig. Future experiments will be designed better to take into account the various influences for better clarify. We have added the references about the utilization of manta in Human and mink.

Point 2: Furthermore, the authors should consider validating some CNVs of interest using qPCR as proof of reliable data analysis. The matter of validation of data analysis should also be considered in the discussion.

Response 2: Thank you for this valuable suggestion. There are no samples in the lab. Due to the impact of the epidemic, it is temporarily impossible to collect samples. If conditions permit, the verification of CNV and the association analysis between phenotype and CNV should be conducted immediately. We will do better in collecting more samples for verification and phenotypes of interests.

Point 3: AWB: explain abbreviation at first use

Response 3: Thank you very much for your suggestion. I felt sorry for this mistake in the preparing the manuscript. I have explain the abbreviation at first use in line 79 and marked in red.

Point 4: Use consistently among text and figures either “AWB” or “Wild” or else

Response 4: Thank you for valuable suggestion. I have changed and used the AWB in the manuscript in line 82, 103, 105, 108, 124, 133, 157, table 2, and 328.

Point 5: Line 189: terms

Response 5: Thank you for your suggestion. I am sorry for this mistake in manuscript. I have added the detail in line 193 and marked with red.

Point 6: Discussion: First paragraph seems to be a leftover from the template. Remove.

Response 6: I would like to express my heartfelt thanks to you for this suggestion. I felt quite sorry for this mistake. I will be more circumspection for preparing manuscript.

Thank you very much for your constructive advices on our paper, which is of great significance to the overall logic and integrity of our manuscript, and also to learn how to write amanuscript better.

Round 2

Reviewer 2 Report

The authors addressed all comments thoroughly. The authors made clear that this study is supposed to be a screening for potentially interesting genomics regions and will investigate the copy numbers in their subsequent work. I accept the manuscript in its current state.